# A Multi-Stage Progressive Pansharpening Network Based on Detail Injection with Redundancy Reduction

**DOI:** 10.3390/s24186039

**Published:** 2024-09-18

**Authors:** Xincan Wen, Hongbing Ma, Liangliang Li

**Affiliations:** 1School of Computer Science and Technology, Xinjiang University, Urumqi 830046, China; 107552201337@stu.xju.edu.cn; 2Key Laboratory of Signal Detection and Processing, Xinjiang University, Urumqi 830046, China; 3Department of Electronic Engineering, Tsinghua University, Beijing 100084, China; 4School of Information and Electronics, Beijing Institute of Technology, Beijing 100081, China; leeliangliang@163.com

**Keywords:** deep learning, pansharpening, feature extraction, image reconstruction, multi-stage

## Abstract

In the field of remote sensing image processing, pansharpening technology stands as a critical advancement. This technology aims to enhance multispectral images that possess low resolution by integrating them with high-spatial-resolution panchromatic images, ultimately producing multispectral images with high resolution that are abundant in both spatial and spectral details. Thus, there remains potential for improving the quality of both the spectral and spatial domains of the fused images based on deep-learning-based pansharpening methods. This work proposes a new method for the task of pansharpening: the Multi-Stage Progressive Pansharpening Network with Detail Injection with Redundancy Reduction Mechanism (MSPPN-DIRRM). This network is divided into three levels, each of which is optimized for the extraction of spectral and spatial data at different scales. Particular spectral feature and spatial detail extraction modules are used at each stage. Moreover, a new image reconstruction module named the DRRM is introduced in this work; it eliminates both spatial and channel redundancy and improves the fusion quality. The effectiveness of the proposed model is further supported by experimental results using both simulated data and real data from the QuickBird, GaoFen1, and WorldView2 satellites; these results show that the proposed model outperforms deep-learning-based methods in both visual and quantitative assessments. Among various evaluation metrics, performance improves by 0.92–18.7% compared to the latest methods.

## 1. Introduction

In the past few decades, the enhancement of remote sensing technologies has impacted several fields by making it convenient to incorporate high-resolution multispectral (HRMS) images. These images are important for instances like detecting land [1], monitoring targets [2], and semantic segmentation [3], among others [4]. However, satellite sensor imaging systems are still limited in their ability to provide remote sensing images with high spatial as well as high spectral resolution. Usually, these systems obtain low-spatial-resolution multispectral (LRMS) images and high-spatial-resolution panchromatic (PAN) images by using different sensors [5]. As a result of these constraints, there has been the development of what is known as pansharpening technology. This technique improves the spatial resolution of multispectral images with the help of panchromatic images and produces high-resolution multispectral images.

Existing pansharpening methodologies are broadly classified into two categories: conventional and deep learning (DL)-based methods. Traditional methods are mainly the component substitution (CS) method [6], the multiresolution analysis (MRA) method [7], and the variational optimization (VO) method [8]. All of these techniques utilize different approaches to solve the problem of increasing the spatial resolution of multispectral images, and they differ in their performance and difficulty.

CS algorithms work in the following way: the MS image is first upsampled and then projected onto a certain space, where the image is separated into spatial and spectral components. The spatial components of this image are then replaced with the panchromatic (PAN) images, which are endowed with more spatial details. The last step is to perform an inverse transformation on the LRMS image to generate the HRMS image. CS algorithms that are worth mentioning are intensity–hue saturation transform (IHS) [9], partial replacement adaptive CS (PRACS) [10], and Brovey [11]. However, these methods generally cause serious spectral distortion in the fused images since the MS spatial bands are replaced by the PAN details without proper reference to compatibility.

Techniques derived from MRA commonly utilize a high-pass filter for extracting detailed information from PAN images. Then, these details are fused into LRMS images to generate HRMS images. Some of the well-known algorithms in this class are the adaptive weighted Laplacian pyramid (AWLP) [12], modulation transfer-function-generalized Laplacian pyramid (MTF_GLP) [13], and induction scaling approach (Indusion) [14], among others. However, these methods have some limitations, such as spatial distortions (e.g., blurring) or artifact formation, which affect the image quality.

VO in the context of remote sensing image fusion that utilizes variational theory as its method is called variational optimization. These methods work on the basis of the smoothness of the image and apply certain conditions on image reconstruction. These constraints form the basis of the formation of an energy functional to be used in solving image processing problems. This process results in the attainment of the final image based on minimizing the value of the generalized function, as stated in [15]. Some of the well-known VO algorithms are P+XS methods [16], Bayesian-based methods [17], and sparse-representation-based methods [18]. However, they also have a significant disadvantage: the time needed for their execution is much longer compared to other methods, which is why their further development and application in high-speed applications is limited.

DL uses neural networks to simulate the learning process of the human brain; it trains models with large amounts of data in order to automatically extract features, recognize patterns, and perform classification and prediction tasks. In recent years, deep learning has been widely applied in the field of image processing, such as SAR image change detection [19], cross-view intelligent person searches [20], landslide extraction [21], shadow detection in remote sensing imagery [22], and building height extraction [23].

Convolutional neural networks (CNNs), as key models in the field of deep learning, are considered one of the most valuable approaches in the pansharpening domain because of their ability to extract features and their effective generalization. Based on the introduction of CNNs in the field of super-resolution reconstruction of remote sensing images [24], Masi et al. [25] proposed the first DL-based pansharpening work with a three-layer convolutional network model (PNN). This model paved the way for other developments in the area of study. After this innovation, the research community came up with better CNN models. For instance, the multi-scale and multi-depth CNN (MSDCNN) [26] is intended for multi-scale and multi-depth feature extraction through networks that operate on shallow and deep information layers. Xu et al. [27] later improved it by adding a residual module to form a residual-network-based PNN (DRPNN). This method also improves detail capture by applying residual learning. Another significant improvement was described in [28], wherein an end-to-end method is used to insert obtained details into the upsampled multispectral images to improve overall image quality. Jin [29] designed a full-depth feature fusion network (FDFNet) consisting of three branches (MS branch, PAN branch, and fusion branch). The MS branch extracts spectral information from MS images; the PAN branch extracts rich texture and detail information from PAN images. The information extracted by these two branches is injected into the fusion branch, achieving full-depth fusion of the network. In addition, Zhang [30] proposed LAGConv as a new convolution kernel that learns the local context with global harmonics to overcome the issue of spatial discontinuity and to balance global and local features. Jun Da et al. [31] introduced the cascaded multi-receptive learning residual block (CML) to learn features from multiple receptive fields in a structured manner in order to improve pansharpening performance. The cross spectral–spatial fusion network (CSSFN) [32] uses a two-stage residual network structure to extract information at different scales from MS and PAN images and uses a cross-spectral–spatial attention block (CSSAB) to interact with spatial information and spectral information, improving the quality of the fusion results.

Despite the significant advancements achieved by deep-learning-based pansharpening methods, there are notable challenges that still need to be addressed:Overlooking of size variability: Some methodologies, exemplified by the works of [27] and [31], operate under the assumption that changes in the sizes of MS and PAN images do not impact the fusion outcomes. This assumption fails to consider the variable information content present in MS and PAN images at different scales, and the networks often utilize simplistic, single-stage fusion. This approach can overlook critical details necessary for high-quality fusion.Undifferentiated feature handling: Techniques such as the one mentioned in [30] process stacked feature maps of MS and PAN images as uniform inputs into the feature extraction network or utilize identical subnetworks for both MS and PAN feature extraction (such as [29]). This method does not account for the inherent differences between spectral and spatial features, potentially leading to a lack of model interpretability and underutilization of the unique characteristics each feature type offers.Overlooking of redundancy: Most existing methods (such as [32]) perform feature fusion after feature extraction, during which the generation of redundant information is inevitable. However, these methods lack consideration for eliminating redundant information, resulting in a decrease in the quality of the fused results.

In response to the above issues, instead of adopting a simple end-to-end single-stage fusion network, we propose a multi-stage fusion pansharpening network based on detail injection with redundancy reduction that can fully extract spectral and spatial information from MS and PAN images at different resolutions. In order to obtain richer spectral information from MS images, we design an efficient channel feature extraction module (CFEM) to model spectral features by learning the relationships between feature map channels. In order to obtain sufficient spatial details from PAN images without introducing additional redundant information, inspired by partial convolution (PConv) [33] and spatial attention (SA) [34], we design a space detail extraction module (SDEM). At the end of each stage, we reconstruct the previously extracted features. Specifically, the fusion features obtained by this module are first subjected to spatial and channel redundancy reduction operations to obtain feature maps with spatial and channel enhancement, respectively. These two sets of feature maps undergo comprehensive interaction followed by sequential operations to reduce spatial and channel redundancy. Experimental results on the datasets collected by the QuickBird (QB), GaoFen1 (GF1), and WorldView2 (WV2) satellites show that our proposed method has a certain level of improvement in performance compared to other methods, and the number of parameters used is relatively small. Our contributions are summarized as follows:A multi-stage progressive pansharpening framework is introduced to address both the spectral and spatial dimensions across varying resolutions in order to incrementally refine the fused image’s spectral and spatial attributes while maintaining stability throughout the pansharpening process.A spatial detail extraction module and a channel feature extraction module are developed to proficiently learn and distinguish between the characteristics of spectral and spatial information. Concurrently, a detail injection technique is employed to enhance the model’s interpretability.The designed DRRM mitigates redundancy within both the spatial and channel dimensions of the feature maps’ post-detail injection, fostering enhanced interaction among diverse information streams. This approach not only enriches the spectral and spatial representations in the fusion outcomes but also effectively addresses the issue of feature redundancy.

## 2. Related Works

### 2.1. Detail Extraction and Injection

The CS method and MRA method are established traditional techniques in the field of pansharpening. The methodology of detail extraction and subsequent injection, as discussed in this paper, draws inspiration from these foundational approaches. The fundamental workflows of both techniques can be delineated into three primary steps: (1) Identifying the discrepancies between the PAN image and a specific image, (2) amplifying these discrepancies by applying a predefined amplification factor, and (3) Infusing the amplified discrepancies into the upscaled LRMS images to derive HRMS images. The mathematical representation of this process is encapsulated in the Equation (Equation 1) below:(1)$$\hat{MS_{b}}=\tilde{MS_{b}}+g_{b}⊙\left(P-X\right)$$

In this formulation, $\hat{MS_{b}}\in R^{M\times N}$, $\tilde{MS_{b}}\in R^{M\times N}$ represent the *b*-th band of the HRMS and LRMS images, respectively, and are both within $R^{M\times N}$. The variable $g_{i}\in R$ denotes the *i*-th injection coefficient, which modulates the details during the injection process. The matrix $P\in R^{M\times N}$ symbolizes the PAN image. In the context of MRA, ***X*** signifies the low-pass-filtered version of the PAN image. Conversely, within the CS framework, ***X*** denotes the intensity component derived from a linear combination of the LRMS channels, which is expressed as follows:(2)$$I=\sum_{b=1}^{B}\omega_{b}\tilde{MS_{b}}$$In (Equation 2), $\omega_{b}$ is the weight of the *b*-th band. Formula (Equation 1) can further be written as Formula (Equation 3):(3)$$\hat{\mathrm{MS}}=\tilde{\mathrm{MS}}+g⊙\left(P^{D}-X^{D}\right)$$

The terms $\hat{\mathrm{MS}}\in R^{M\times N\times C}$ and $\tilde{\mathrm{MS}}\in R^{M\times N\times C}$ consist of independent bands $\hat{MS_{b}}\in R^{M\times N}$ and $\tilde{MS_{b}}\in R^{M\times N}$ for each *b* = 1, 2, ⋯, *C*, where *C* represents the number of channels in the LRMS images. The matrices $P^{D}\in R^{M\times N\times C}$ and $X^{D}\in R^{M\times N\times C}$ are derived by duplicating the PAN image and the *X* image, respectively, across *C* channel dimensions. The vector $g\in R^{C}$ corresponds to the set of correlation coefficients $g_{i}$ as specified in Formula (Equation 1), where ⊙ denotes the element-wise multiplication of each element g with the corresponding elements in the difference $\left(P^{D}-X^{D}\right)$ across each band.

Both CS and MRA methods utilize common linear injection models, yet they diverge in their initial assumptions. In the CS approach, it is preliminarily presumed that the spectral model used projects the MS images onto PAN images. Conversely, the MRA method typically assumes the configuration of the high-pass filter applied to PAN images; inappropriate assumptions in either method can lead to suboptimal image fusion outcomes. Furthermore, both methods require estimation of the injection coefficients $g_{b}$. To address this issue, Deng [35] implemented a method in which the PAN image, expended to $P^{D}$, is subtracted from the upsampled LRMS image, $\tilde{\mathrm{MS}}$. This approach effectively separates the high-resolution details captured by the PAN image from the multispectral data. The resulting difference, $\left(P^{D}-\tilde{\mathrm{MS}}\right)$, is then processed through a deep convolutional neural network (DCNN) to extract the requisite details for injection $\tilde{\mathrm{MS}}$. This process is mathematically articulated in Formula (Equation 4):(4)$$\hat{\mathrm{MS}}=\tilde{\mathrm{MS}}+f_{\theta s}\left(P^{D}-\tilde{\mathrm{MS}}\right)$$
where $f_{\theta s}$ is a nonlinear mapping operation with its associated parameters.

Additionally, the concept of detail injection has gained extensive application across various fields in recent years. For instance, Liu [36] introduced a spatiotemporal fusion model predicated on detail injection. This model features a three-branch detail injection (TDI) module that is designed to transfer unaltered details extracted from neighboring high-resolution images into the target image. This approach effectively preserves the abrupt change information captured from the coarser target image, thereby facilitating the reconstruction of superior-quality high-resolution target images.

### 2.2. Multi-Stage Progressive Pansharpening

In remote sensing research, the images captured by sensors encompass varying information across distinct scales, such as diverse surface morphologies at different spatial scales. In super-resolution (SR) reconstruction, there are already methods that use multi-stage progressive network structures that consider scale factors to improve the quality of generated images. In SR reconstruction, the most typical multi-stage progressive structure is the LapSRN model [37]. It processes low-resolution images through a network that incrementally constructs high-resolution images across multiple scales.

The parallels between pansharpening and SR reconstruction center on the generation of high-resolution images from low-resolution observations [38]. Therefore, some pansharpening methods also adopt a multi-stage progressive structure. The SRPPNN framework [39] initially introduces a progressive pansharpening architecture, which segregates the network into two subnetworks. Each subnetwork is tailored to extract spectral and spatial information at different scales and applies 2× upsampling to the multispectral input. Li et al. [40] employ a three-stage progressive network structure to fully exploit the spatial and spectral information of images at different scales. As the spatial scale gradually increases, the spectral and spatial features extracted by the network are also gradually enhanced. Similarly, the DPFN [41] and P2Sharpen [42] models have adopted analogous structures to amplify the pansharpening capabilities of their respective frameworks.

### 2.3. Attention Mechanism

Due to the characteristics of the human visual system, key areas in complex scenes are easily noticed and understood by people. Inspired by this, computer vision typically uses attention mechanisms that mimic the human visual system to adaptively capture valuable information for the required task.

In recent years, many deep-learning-based pansharpening methods have introduced attention mechanisms in the design of network models: for example, the CSSFN mentioned in the ’Introduction’. In addition, Diao [43] proposed a pansharpening network with triple attention and designed corresponding attention modules for spectral information, spatial information, and information stacked together to highlight feature maps containing important information. Jiang [44] designed a spatial global self-attention module based on the similarity of spatial structures in remote sensing images that enhances the feature representation of fusion results by selectively enhancing texture details. Yang [45] designed spatial spectral joint-attention in the process of image reconstruction to guide the network to recover spectral and spatial information from feature maps extracted from MS and PAN images separately.

## 3. Proposed Method

### 3.1. Overview of the Overall Framework

The architecture of the proposed method is depicted in Figure 1 and comprises three similar subnetwork structures. Each subnetwork is designed to extract spectral and spatial information at different scales using the CFEM and the SDEM, respectively. According to the Wald [46] protocol, the multispectral (MS) images and panchromatic (PAN) images, which are processed separately, are treated as original images within this network model. The term ${\mathrm{Fusion}}_{\left(i\right)}$ denotes the fusion outcome of the *i*-th stage. The symbols X↑r and X↓r indicate the operations of upsampling by a factor of r and downsampling by a factor of *r*, respectively, to X. The notation $X\left(\cdot \right)$ refers to processing by a specific module that we designed, with all subsequent occurrences $X\left(\cdot \right)$ sharing the same meaning. CFEM mainly captures spectral information between channels, while LRMS, ${\mathrm{Fusion}}_{\left(1\right)}↑2$, and ${\mathrm{Fusion}}_{\left(2\right)}↑2$ are all multi-band images. These multi-band images have rich spectral information between different channels. At the same time, in order to maintain the size of the feature maps processed in each stage and obtain spectral information of different sizes, at the first, second, and third stages, the inputs for the CFEM are LRMS, ${\mathrm{Fusion}}_{\left(1\right)}↑2$, and ${\mathrm{Fusion}}_{\left(2\right)}↑2$, respectively. To mitigate the issue of spectral distortion that is often encountered in detail-injection-based models as discussed in DFPN [41], according to the idea of detail feature extraction and injection from Formula (4) in Part A of a related work, the inputs for the SDEM at these stages are $\left(\mathrm{PAN}↓4-\mathrm{LRMS}\right)$, $\left(\mathrm{PAN}↓2-{\mathrm{Fusion}}_{\left(1\right)}↑2\right)$, and $\left(\mathrm{PAN}-{\mathrm{Fusion}}_{\left(2\right)}↑2\right)$, respectively. This strategy ensures that spectral information is effectively integrated into the detail extraction process while obtaining source detail information of different sizes. In addition, in each stage, the output detail information of SDEM is injected into the spectral features extracted by CFEM. Additionally, each subnetwork utilizes a DRRM to eliminate redundant information and enhance the depiction of spectral and spatial information in the fused images. Furthermore, to fully capitalize on the information available in the source images, each subnetwork receives LRMS and PAN images that have been either upsampled or downsampled to match the scale processed by the respective subnetwork. In summary, the operations of our proposed network model are formally expressed by Formulas (Equation 5)–(Equation 7):(5)$${\mathrm{Fusion}}_{\left(1\right)}=\mathrm{DRRM}(\mathrm{CFEM}(\mathrm{LRMS})+\mathrm{SDEM}\left(\mathrm{PAN}↓4-\mathrm{LRMS}\right))$$
(6)$${\mathrm{Fusion}}_{\left(2\right)}=\mathrm{DRRM}(\mathrm{CFEM}\left({\mathrm{Fusion}}_{\left(1\right)}↑2\right)+\mathrm{SDEM}(\mathrm{PAN}↓2-\mathrm{LRMS}↑2))$$
(7)$${\mathrm{Fusion}}_{\left(3\right)}=\mathrm{DRRM}\left(\mathrm{CFEM}\left({\mathrm{Fusion}}_{\left(2\right)}↑2\right)+\mathrm{SDEM}\left(\mathrm{PAN}-\mathrm{LRMS}↑4\right)\right)$$

The following subsections provide a detailed introduction to the CFEM, SDEM, and DRRM.

### 3.2. Channel Feature Extraction Module

Spectral information in multispectral images typically arises from the interdependent relationships among various channels. Contemporary attention mechanisms enhance performance by exploring more complex inter-channel dependencies or by incorporating additional spatial attention mechanisms, as seen in [34,47]. However, Wang [48] has shown through empirical research that the dimensionality reduction employed in the well-known self-attention (SE) [49] method can diminish the accuracy of predictions, suggesting that capturing dependencies across all channels may not always be necessary. To avoid the loss of spectral information caused by channel dimensionality reduction and enhance the information interaction between different channels, in the spectral extraction module shown in Figure 2, we introduce an efficient attention mechanism [48] to map the band relationships in multispectral images onto feature maps.

By determining the weights of each feature map, we strive to capture spectral information as precisely as possible by facilitating the interaction of spectral data across different channels and minimizing spectral information loss. The channel feature extraction module processes the input feature map, denoted as $X_{0}$, through a convolutional layer equipped with a 3 × 3 filter and a ReLU activation function, yielding $X_{1}$. Subsequently, $X_{1}$ is subjected to global average pooling (GAP) without dimensionality reduction, leading to the aggregation of channel features. At this juncture, a one-dimensional convolution of size k is employed to enable cross-channel information exchange. The size of k is adaptively determined by the number of channels, as outlined in Formula (Equation 8):(8)$$k=\psi \left(C\right)={\left|\log_{2}\left(C\right)+1\right|}_{\mathrm{odd}}$$The term ${\left|t\right|}_{\mathrm{odd}}$ represents the odd integer closest to t. Furthermore, the variable k denotes that k adjacent channels to a specific channel contribute to predicting that channel’s weight, which is represented ψ(C). Ultimately, the weight map produced via the sigmoid activation function is applied to $X_{1}$ through channel-wise multiplication, resulting in $X_{2}$. Generally, enhancing the width and depth of a network is an effective means to improve its performance. Deep networks typically outperform shallow ones; however, merely increasing the depth of a network may induce issues such as gradient dispersion or gradient explosion [50]. To address these challenges and maximize the utilization of spectral information at the input, this module incorporates a skip connection strategy. The output from the entire module is succinctly articulated in Formula (Equation 9):(9)$$X_{3}=X_{1}+X_{2}$$

### 3.3. Spatial Detail Extraction Module

In contrast to the spectral extraction module, the spatial detail extraction module is designed to capture spatial information, as depicted in Figure 3a. Similar to the process illustrated in Figure 2, this module also utilizes a convolutional layer with a 3 × 3 kernel and a ReLU activation function and incorporates a skip connection strategy. References [51,52] have indicated that feature maps located in different channels exhibit high levels of similarity. To diminish information redundancy across these channels and to effectively extract spatial information, we implement PConv [33]. PConv differs from standard convolution, which processes features across all channels; instead, PConv selectively extracts spatial features from specific input channels, enhancing efficiency. Following PConv, a point-wise convolution (PWConv) is added to fully capitalize on the spatial information across all channels and to finalize the preliminary extraction of spatial information. Assuming the input signal is $F_{0}$, the preliminarily extracted feature map can be expressed using Formula (Equation 10):(10)F1=ReLUConv3×3F0+PWConvPConvReLUConv3×3F0To simplify the representation, ReLuConv3×3(·) is abbreviated as RC(·). To further refine the extraction of spatial details and enhance the focus on positional information within images, we incorporate a spatial attention mechanism [34], as depicted in Figure 3b. Specifically, for the obtained feature maps $F_{1}\in R^{C\times H\times W}$, both max pooling and average pooling operations are conducted along the channel dimension. This results in $F_{1M}\in R^{1\times H\times W}$ (representing the maximum pooling feature of the channel) and $F_{1A}\in R^{1\times H\times W}$ (representing the average pooling feature of the channel). These two feature maps, $F_{1M}$ and $F_{1A}$, are then concatenated and subsequently passed through a convolutional layer followed by a sigmoid activation function to produce a spatial weight map $W\in R^{1\times H\times W}$. This procedure is mathematically formulated in Equation (Equation 11):(11)$$W=\sigma (Conv_{2,1}\left(Concate\left[F_{1M},F_{1A}\right]\right))$$
In this context, σ denotes the sigmoid activation function, and $Conv_{2,1}\left(\cdot \right)$ represents a convolutional layer that has two input channels and one output channel. After obtaining the spatial weight map W, each element of W is multiplied element-wise with the corresponding elements of the feature map $F_{1}$ across each channel to integrate spatial information effectively. The output from this module is succinctly captured in Equation (Equation 12):(12)$$F_{2}=F_{1}+RC\left(SA\left(RC\left(F_{1}\right)\right)\right)$$

### 3.4. Dual Redundancy Reduction Module

Numerous studies [53,54] have demonstrated that DCNNs often harbor a considerable amount of redundant information. As discussed earlier in this article, both the CFEM and SDEM incorporate skip connections to preserve original image information, which inadvertently introduces a significant amount of redundancy. When the outputs of these two branches are combined, redundancy inevitably arises. Inspired by SCconv [55], which implements a novel CNN approach to simultaneously reduce spatial and channel redundancy, we design the DRRM, as depicted in Figure 4. The DRRM is structured in three stages: initially, channel and spatial information are reconstructed separately through the channel redundancy reduction module (CRRM) and the spatial redundancy reduction module (SRRM), respectively; this is followed by the information exchange module (IEM), wherein channel and spatial information interact comprehensively; finally, as spatial redundancy persists in the feature maps processed by the CRRM and channel redundancy persists in those processed by the SRRM, sequential processing of the SRRM and CRRM refines the feature maps in both the channel and spatial dimensions. In the following sections, we provide detailed descriptions of the SRRM, CRRM, and IEM.

As illustrated in Figure 5, the spatial redundancy reduction module (SRRM) involves both separation and reconstruction operations. The separation operation categorizes feature maps into two types: those with rich spatial information and those with sparse spatial information. Specifically, for the feature maps obtained after feature extraction, represented by $X\in R^{N\times C\times H\times W}$, we initially utilize the scaling factor in the group normalization (GN) layer [56] to assess the relative importance of information content across different feature maps. This is followed by weight normalization to derive normalized correlation weights ($W_{r}$) that signify the importance of different feature maps. Subsequently, a threshold of 0.5 is applied in a gate operation to isolate the information weight $W_{1}$. This process is mathematically detailed in Equation (Equation 13):(13)$$W_{r}=\left\{\begin{array}{cc} 1, & w_{r}\geq \frac{1}{2},\\ 0, & \mathrm{otherwise}. \end{array}\right.$$

In Equation (Equation 13), the feature maps for which $w_{r}\geq \frac{1}{2}$ contain rich spatial information, whereas those for which $w_{r}<\frac{1}{2}$ are characterized by sparse spatial information. Additionally, executing the inverse gating operation relative to the initial procedure results in a non-informational weight $W_{2}$. Subsequently, the input feature map X is multiplied by $W_{1}$ and $W_{2}$, respectively yielding $X_{1}^{W}$, which exhibits strong spatial expressiveness, and $X_{2}^{W}$, which contains minimal spatial information.

For the reconstruction phase, a cross-reconstruction operation is employed between $X_{1}^{W}$ and $X_{2}^{W}$ with the aim of fully integrating and enhancing the information exchange between these two components. The final step involves concatenating the results. The specific process is described by the following group of equations, in which ∪ denotes the concatenation operation, $X_{11}^{W}$ and $X_{12}^{W}$ are two-part feature maps split from $X_{1}^{W}$, and $X_{21}^{W}$ and $X_{22}^{W}$ are two-part feature maps split from $X_{2}^{W}$:(14)$$\left\{\begin{array}{c} X_{11}^{W}⊕X_{22}^{W}=X^{W1}\\ X_{21}^{W}⊕X_{12}^{W}=X^{W2}\\ X^{W1}\cup X^{W2}=X^{W} \end{array}\right.$$

The channel redundancy reduction module (CRRM) is designed to perform segmentation, transformation, and fusion operations on the input feature map, as depicted in Figure 6. The segmentation process involves dividing the channel into two groups with channel counts of αC and (1−α)C using the partitioning factor α, followed by applying 1 × 1 convolution kernels to compress the channels of the two groups of feature maps, respectively, resulting in $X_{\mathrm{up}}$ and $X_{low}$. The transformation process uses $X_{\mathrm{up}}$ as the input for “rich feature extraction”; $X_{\mathrm{up}}$ performs group convolution operations and pointwise convolution operations separately and then combines them to produce $Y_{1}$. Simultaneously, $X_{low}$ serves as a supplementary input to the “rich feature extraction”; it undergoes pointwise convolution operations, and the results are concatenated with $X_{low}$.

The fusion operation is more complex than simple addition or concatenation. It begins with performing global average pooling operations on the outputs of the two branches, $Y_{1}$ and $Y_{2}$, to obtain global channel descriptors $S_{1}$ and $S_{2}$ for the upper and lower branches, respectively. A softmax operation is then applied to $S_{1}$ and $S_{2}$ to obtain the vectors $\beta_{1},\beta_{2}\in R^{C}$, which represent the importance of each branch. Finally, $Y_{1}$ and $Y_{2}$ are fused channel-by-channel using $\beta_{1}$ and $\beta_{2}$, as shown in the equation:(15)$$Y=\beta_{1}Y_{1}+\beta_{2}Y_{2}$$

The spatial redundancy reduction module (SRRM) and channel redundancy reduction module (CRRM) each effectively refine features within their respective domains—spatial and channel—but traditionally lack interaction between each other. The information exchange module (IEM) addresses this gap efficiently. As depicted in Figure 4, networks from different branches are organized according to channel dimensions, facilitating comprehensive interaction between diverse types of information. This arrangement enhances the integration of spatial- and channel-specific details, enabling a more cohesive and informed fusion process.

### 3.5. Design of the Loss Function

In the development of loss functions, the predominant approach involves employing the $\ell_{1}$ loss function, which focuses on minimizing the absolute pixel differences between the merged image and the reference image, ensuring the fusion outcome closely resembles the original image [57]. Nonetheless, this method primarily enhances image fusion from an objective standpoint, neglecting human subjective perception. To address this, the current paper incorporates MS-SSIM [58]: a metric capable of flexibly and precisely comparing images in terms of brightness, structure, and contrast to better reflect human visual assessment. This loss function has been successfully utilized in studies like [59] and has yielded positive results. The MS-SSIM calculation formula is as follows:(16)$$L_{MS-SSIM}={\left[L_{M}\left(x,y\right)\right]}^{\alpha_{M}}\prod_{i=1}^{M}{\left[C_{i}\left(x,y\right)\right]}^{\beta_{i}}{\left[S_{i}\left(x,y\right)\right]}^{\gamma_{i}}$$
where M is the number of multiple scales, usually five. $L_{M}\left(x,y\right)$, $C_{i}\left(x,y\right)$ and $S_{i}\left(x,y\right)$ represent brightness changes at the M-th scale, contrast changes at the i-th scale, and structural changes at the i-th scale, respectively; $\alpha_{M}$, $\beta_{i}$, and $\gamma_{i}$ represent the corresponding equilibrium parameters.

In summary, the formula for the loss function used in this article is as follows:(17)$$L=\alpha L_{1}+\left(1-\alpha \right)\left(1-L_{MS-SSIM}\right)$$
where α is the equilibrium parameter, and, according to Zhao’s previous conclusion [58], it is set to 0.16 to ensure the best performance.

## 4. Experimental Results

### 4.1. Datasets

To assess the performance of the proposed method, this study conducted validations on both simulated and real datasets from the QB, GF1, and WV2 satellites. The LRMS images from QB and GaoFen1 contain four bands, while the LRMS images from WV2 contain eight bands. The PAN images from the three satellite datasets are all single-band images. Specifically, the spatial resolutions for the low-resolution multispectral (LRMS) and PAN images in the QB dataset are 2.44 m and 0.61 m, respectively, while for the GaoFen1 dataset, they are 8 m and 2 m, respectively. As for WV2, they are 2 m and 0.5 m, respectively. More specific information and the acquisition methods for these three datasets are detailed in [60]. This evaluation includes tests with both simulated and real data to objectively verify the method’s effectiveness. Although there is no directly usable simulation dataset for the simulated data evaluation, training pansharpening networks based on CNN is conducted on simulated datasets. Therefore, we adopt the Wald protocol. This involves applying modulation transfer function (MTF) [13] filtering to the original LRMS and PAN images, followed by 4× downsampling. After this process, we obtain 3000 pairs of simulated images from the QB dataset, 2460 pairs of simulated images from the GF1 dataset, and 3000 pairs of simulated images from the WV2 dataset. The original LRMS image serves as the reference image. For each simulated dataset, the distribution is 80% for training, 10% for validation, and 10% for testing. The sizes of the LRMS and PAN images used are 64 × 64 × 4 (64 × 64 × 8) and 256 × 256, respectively. For the real data evaluation, the original LRMS and PAN images (with sizes of 256 × 256 × 4 (256 × 256 × 8) and 1024 × 1024, respectively) are used directly as inputs of networks for testing (the original image pairs are only used as the testing dataset), with each satellite’s testing dataset containing the same number of image pairs as the simulated data evaluation.

### 4.2. Evaluation Indicators and Comparison Methods

In the simulated dataset, we employ several evaluation metrics to assess image quality and accuracy. These metrics include ERGAS (erreur relative globale adimensionnelle de synthese) [61], RMSE (root mean square error), RASE (relative average spectral error) [61], UIQI (universal image quality index) [61], SAM (spectral angle mapping) [61], SCC (structural correlation coefficient) [62], and Q4 [61]. It is important to note that higher values of UIQI, SCC, and Q4 indicate better quality, whereas lower values of ERGAS, RMSE, RASE, and SAM signify superior performance. For the real dataset, the evaluation focuses on full-resolution performance using metrics such as the QNR (quality without reference), spectral distortion index ($D_{\lambda }$), and spatial distortion index ($D_{s}$) [63]. A larger QNR value denotes higher quality, while smaller $D_{\lambda }$ and $D_{s}$ values indicate less distortion.

This article compares the effectiveness of our proposed method with several classic techniques including Brovey [11], MTF_GLP [13], Indusion [14], DRPNN [27], FusionNet [35], FDFNet [29], LAGConv [30], CML [31], and CSSFN [32]. Among these, Brovey, Indusion, and MTF_GLP are traditional methods, whereas the others are DL-based approaches.

### 4.3. Experimental Details

All traditional methods were executed using MATLAB R2021b on a computer equipped with an Intel(R) Core(TM) i9-14900K processor and 16GB of RAM. All DL-based network training and testing were conducted using the PyTorch framework with acceleration provided by a GeForce RTX 3090 graphics card. For the proposed method, the model was trained using the Adam optimizer over 500 training rounds with an initial learning rate of 1 × 10^−4^. The learning rate decayed to 10% of its original value every 100 training rounds. Additionally, the batch size was set to four.

### 4.4. Experimental Results on Simulated Datasets

Figure 7 presents the fusion results of various methods on the QB simulation dataset, with a focus on the enlarged area of the roof of a blue building located in the upper left corner. Figure 8 illustrates the absolute errors between the fusion results of all methods and the reference image, with the magnitude of errors correlated to the color bar shown in Figure 8. Notably, the fusion results of the FusionNet and FDFNet methods display significant spectral distortion. Spatially, the Brovey method excessively enhances texture details and overly emphasizes the roof edges. The fusion result from the Indusion method has some spatial ambiguity on the left edge of the building roof. The fusion result from the CML method shows artifacts, and the DRPNN method’s fusion result has smoothed roof edges. Visually, the fused images from the Hours, LAGConv, and CSSFN methods are closest to the reference image, whereas other methods show some level of distortion in both spectral and spatial information. As further illustrated by Figure 8, the fusion image from the Brovey method exhibits the most severe overall distortion, and the FusionNet method shows significant local distortion. Table 1 provides quantitative results of the fused images generated using different methods on the QB simulation dataset; the best and suboptimal values are highlighted in bold and underlined, respectively (as noted in subsequent tables). From Figure 7 and Figure 8 and Table 1, it is evident that compared to other methods, our proposed method retains the most information in both the spectral and spatial dimensions in the fusion results and shows minimal loss.

Figure 9 presents the fusion results of various methods on the GaoFen1 simulation dataset. Aside from the MTF_GLP method, which causes notable spectral distortion in the fused images, it is challenging to discern significant differences directly in the fusion results of the methods. Therefore, to facilitate detailed comparisons, a small portion of the mountain texture within the overall mountain range is magnified. From a spectral perspective, the fusion images from the Brovey method and FusionNet method exhibit the most substantial color deviation from the reference image. In the fusion image from the FDFNet method, the greenery surrounding mountain crevices appears faded. Meanwhile, the fusion results of other methods manage to preserve spectral information effectively. From a spatial perspective, the Brovey method tends to overly enhance edge details in the fused images. The images from the DRPNN and the Induction method demonstrate a noticeable blurring effect visually. Using the CSSFN method results in smoothed edges around the black ellipse. Intuitively, the fusion results from the CML and the LAGConv methods and our method closely match the reference image. However, as observed in Figure 10 (similarly to Figure 8), compared to these methods, the deviation between the fused image and the reference image using our proposed method is closer to zero. Moreover, the corresponding quantitative analysis detailed in Table 2 indicates that our proposed method performs well and suggests that it preserves the most spectral and spatial information among the evaluated methods.

Figure 11 and Figure 12 show the fusion results of various methods on the simulated dataset of WV2 and their residual maps with the reference image, respectively. In Figure 11, whether analyzed from a global or local perspective, the deep-learning-based method is significantly superior to traditional methods. Specifically, the fusion results of the Brovey method exhibit overall spectral distortion, while the fusion results of the MTF_GLP method exhibit some spectral distortion, and the fusion results of the Infusion method exhibit significant blurriness. From analysis of the enlarged area, the fusion results of DRPNN, FusionNet, and FDFNet do not restore the yellow soil around the green vegetation. The fusion results based on other deep learning methods perform very close to the reference image in the enlarged area, but the edges of the blue thin-bar-shaped part (corresponding to the right side of the red box in the lower left corner of the fusion result) in the fusion results of the LAGConv, CML, and CSSFN methods are not as clear as the fusion result of our proposed method. From Figure 12 (similarly to Figure 8), it can also be seen that our proposed method produces the smallest residual compared to other methods. The quantitative analysis in Table 3 also demonstrates the excellent performance of our proposed method.

### 4.5. Experimental Results on Real Datasets

Figure 13 displays the fusion results of various methods on the real dataset of QB, focusing on a park area within a city. Even though a reference image is not available, it is possible to evaluate the results based on the spectral information from the MS images and the spatial details from the PAN images. Observing the overall color of each image, the fused images produced by the DRPNN, FusionNet, MTF_GLP, and FDFNet methods appear darker. Focusing on the spectral analysis of the magnified area, the paths and roofs between the two lawns in the park are expected to be red. However, the fusion result from the DRPNN method leans towards yellow. The fusion result from the Brovey method shows excessive diffusion of red from the roofs and streets into the surrounding white areas of the lawn, with the LAGConv method displaying a similar, albeit milder, phenomenon. From a spatial perspective, in the enlarged area, the lawn edges in the images from CML and FusionNet are somewhat blurry. In contrast, our proposed method manages to retain both spectral and spatial information effectively in both the overall image and the detailed areas. Table 4 provides a quantitative analysis of the fusion results using different methods on the QB real dataset. Considering both the qualitative and quantitative assessments, the fusion effect of our proposed method proves to be superior to the competing methods.

Figure 14 displays the fusion outcomes of various methods applied to the real dataset from GaoFen1. Overall, methods such as Brovey, MTF_GLP, and FusionNet demonstrate significant deviations from the MS spectral information. Additionally, regarding color reproduction on main roads, the FDFNet and DRPNN methods exhibit lighter colors, whereas other methods more effectively preserve spectral information on roofs. The results from CSSFN are comparable in both the spectral and spatial dimensions: preserving the majority of spectral information but with some blurriness at the edges. In the fusion results obtained using the Infusion method, evidence of color dispersion is observed in the surrounding areas in the images. CML, LAGConv, and our proposed method retain almost complete spectral information, with our proposed method displaying slightly better edge clarity in the fusion results than the CML and LAGConv methods and thus achieving the most effective fusion outcome. Table 5 presents the quality index values of fused images using the different methods on the GF1 real dataset. According to the table, the best $D_{\lambda }$ is achieved by CML, while the highest $D_{s}$ and QNR are attained by our proposed method, highlighting its superiority in maintaining high-quality fusion results.

Figure 15 shows the fusion results of various methods on the real dataset of WV2. Like Figure 13 and Figure 14, there are no reference images in the real dataset. Spatial information refers to PAN images, while spectral information refers to MS images. Clearly, the color in Brovey’s fusion result is overall darker. The following analysis is based on the enlarged regions. The spectral information of buildings in the fusion results of DRPNN and FusionNet is severely distorted and does not match the corresponding spectral information in the MS images. The spectral information in the fusion results of MTF_GLP also differs to some extent from the corresponding spectral information in the MS images. The contour of the building in the fusion result of Infusion is blurred, while the spectral and spatial information of the other methods is well preserved. However, the fusion result of our proposed method is a more vivid fusion image, and Table 6 also shows that the proposed method has certain advantages overall.

### 4.6. Analysis of Ablation Experiment Results

As detailed in Table 7, this paper conducted ablation studies using the QB simulation dataset to evaluate the individual contributions of each module within the proposed method. Comparative experiments were performed by omitting various modules and their combinations to assess their impact. The notation “w/o” denotes the absence of a specific module, while “none_modules” indicates the complete absence of all modules. Figure 16 displays the fusion results after the removal of certain modules. Analysis of Figure 16 reveals that omitting either the CFEM or SDEM alone results in weaker preservation of spatial and spectral details compared to the full model. When the DRRM is omitted, spectral information is severely distorted and spatial details appear particularly blurry. When two modules are omitted, the quality of the fused image is the worst when (CFEM + DRRM) are not used. When (CFEM + SDEM) are not used, the performance is even better than when only the DRRM is removed. This further proves that the DRRM can improve the quality of the fusion results and has the greatest impact on the performance of our proposed model. Figure 17 provides residual plots comparing the outcomes of the experiments with different module omissions against the reference image, demonstrating that the complete model achieves superior fusion performance. Table 7 also illustrates the above conclusions. Furthermore, Table 7 confirms that all metrics of the complete structure are optimal.

### 4.7. Analysis of Experimental Results on Network Structure

To evaluate the effectiveness of the overall network structure, this section develops different network configurations using multiple stages, a cascading stacking strategy (non-detail injection strategy), and the direct injection of PAN images. It then examines the distinctions in their fusion outcomes compared to our proposed original network structure. Figure 18 displays the output results of the various network structures along with their residual plots relative to the reference image. Figure 18d–f illustrate that the output results within the network are progressively enhanced with the injection of information at various scales. Figure 18b indicates that the fusion effects using the cascading stacking strategy are inferior to those achieved with the detail injection method: both in terms of spectral quality and detail resolution. Figure 18c reveals that, as noted by BDPF, the detail injection approach lacks sufficient spectral information injection, leading to some spectral distortion. Specifically, the enlarged area of the windmill exhibits a blue coloration, which is absent in the reference image. The experimental findings and the quantitative analysis presented in Table 8 affirm the effectiveness and soundness of the overall model structure.

### 4.8. Parameter and Running Time Analyses

To enable a more thorough analysis, this article compares the parameter counts of various deep learning methods that have been used previously, as detailed in Table 9. Running time refers to the time it takes for each method to run on a QB simulation test dataset (300 identical images of size 256 × 256 × 4). The measurement of the numbers of parameters and running times for all methods was conducted on the same NVIDIA GeForce RTX 3090 GPU, and all other experimental conditions were the same. It is evident that LAGConv is more compact than our proposed model. As a tradeoff for its larger size, our model outperforms LAGConv in terms of image fusion effectiveness. When compared to other methods, CSSFN and DRPNN involve a significantly higher number of parameters that require training, necessitating substantial datasets for effective training. The proposed model comprises the CFEM, the SDEM, and the DRRM. Each module within our model has been strategically designed to minimize the parameter count, enabling effective training even with limited data samples. This design approach underscores the versatility and adaptability of our proposed model.In addition, the running time of the proposed method is relatively short, and it is somewhat competitive compared to other methods.

## 5. Conclusions

The proposed remote sensing image pansharpening network is a novel one based on a three-stage progressive fusion network. In each stage, the output image is upsampled to twice the size of the input image and a channel attention mechanism is incorporated to enhance spectral features. Additionally, an SDEM is employed to precisely extract spatial details from the PAN image. At the end of each stage, there is information reconstruction by means of a DRRM that enhances the features that were obtained in the prior stage. A comparative analysis with state-of-the-art methods and experiments conducted on both real and synthetic datasets demonstrate that the proposed method surpasses its competitors in terms of subjective and objective evaluation criteria while utilizing fewer parameters. Moreover, the results of ablation studies also support the importance of the channel feature extraction module (CFEM), the SDEM, and the DRRM; on the other hand, extra experiments on the network architecture also prove the advantages of the multi-stage approach.

For future work, the network model that is proposed in this study will be applied to other remote sensing areas like object detection and land classification. Modifications to the proposed network structure will be done in order to fit the requirements of the system. Moreover, it is also observed that the network model has shown superb performance on the simulated dataset, but there is scope for improvement when dealing with real datasets. In the future, research and development will remain devoted to these areas to improve the model’s efficiency. In addition, in order to save computing resources, we will attempt to design new models that can train on datasets from multiple satellites that have been combined into a mixed dataset.

## Figures and Tables

**Figure 1 sensors-24-06039-f001:**
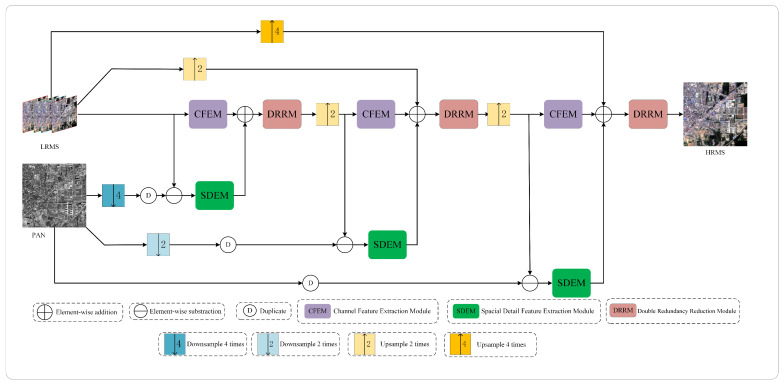
The architecture of the proposed method.

**Figure 2 sensors-24-06039-f002:**
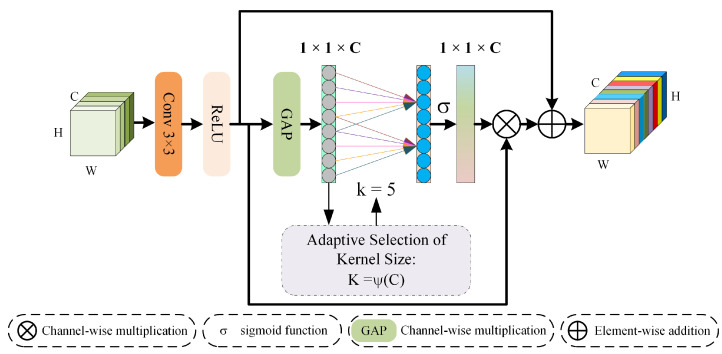
The architecture of the CFEM.

**Figure 3 sensors-24-06039-f003:**
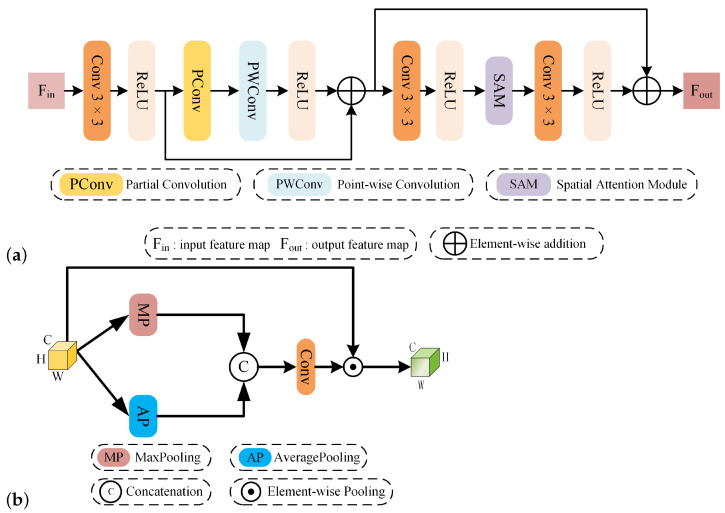
(**a**) The architecture of the overall SDEM. (**b**) The architecture of spatial attention.

**Figure 4 sensors-24-06039-f004:**
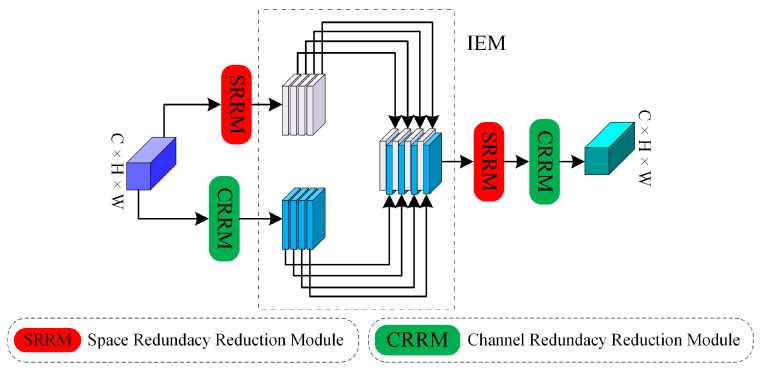
The architecture of the overall DRRM.

**Figure 5 sensors-24-06039-f005:**
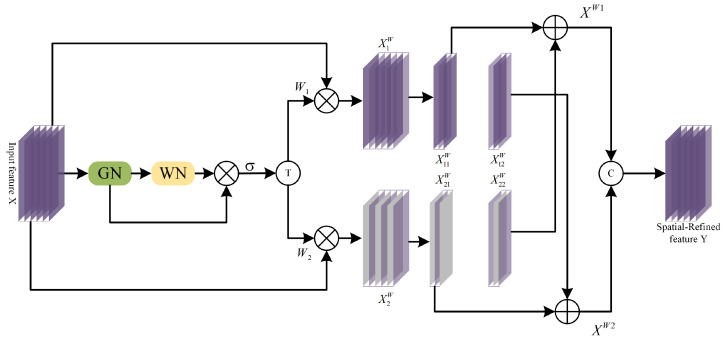
The architecture of the SRRM.

**Figure 6 sensors-24-06039-f006:**
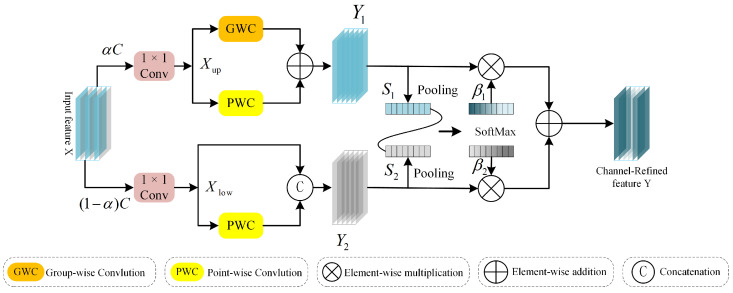
The architecture of the CRRM.

**Figure 7 sensors-24-06039-f007:**
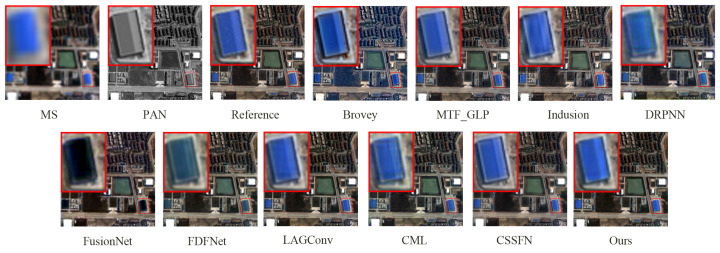
The fusion results of various methods on the QB simulation dataset.

**Figure 8 sensors-24-06039-f008:**
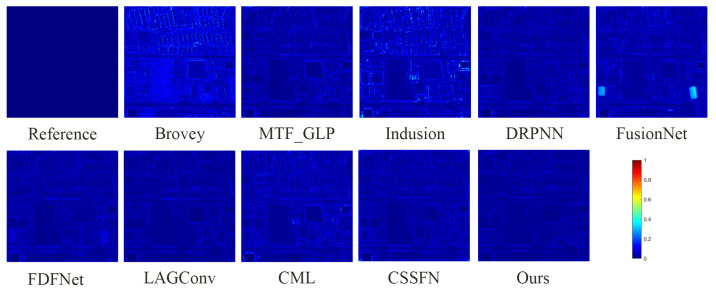
The absolute error maps between the fusion results of all methods and the reference image on the QB simulation dataset.

**Figure 9 sensors-24-06039-f009:**
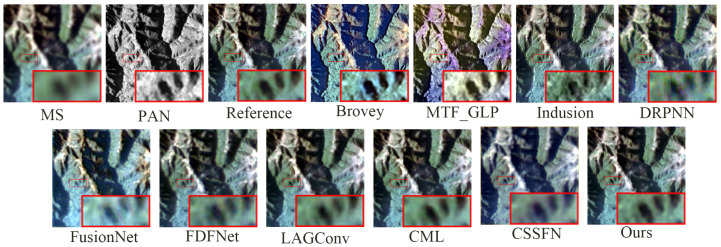
The fusion results of various methods on the GaoFen1 simulation dataset.

**Figure 10 sensors-24-06039-f010:**
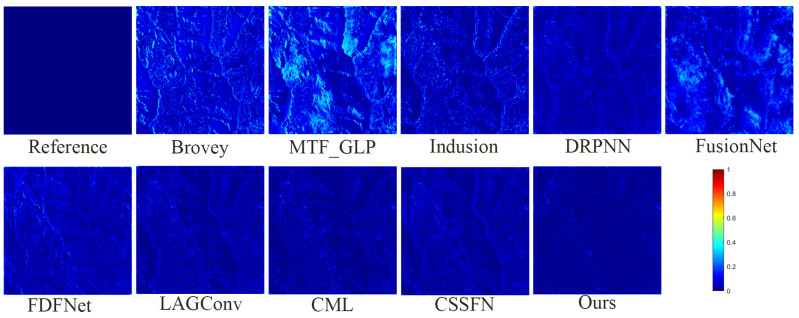
The absolute error maps between the fusion results of all methods and the reference image on the GaoFen1 simulation dataset.

**Figure 11 sensors-24-06039-f011:**
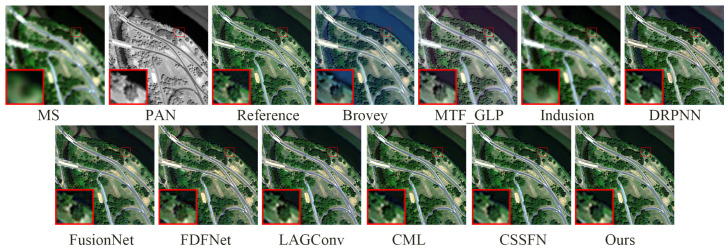
The fusion results of various methods on the WV2 simulation dataset.

**Figure 12 sensors-24-06039-f012:**
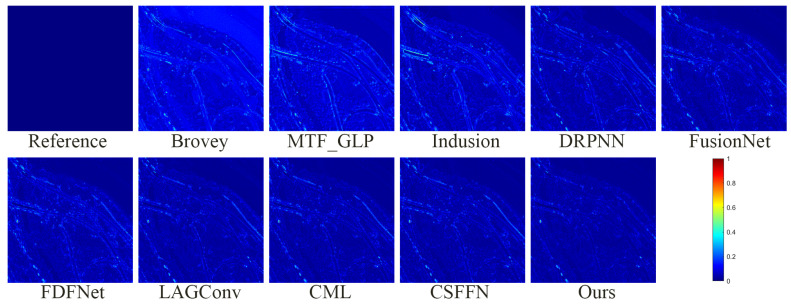
The absolute error maps between the fusion results of all methods and the reference image on the WV2 simulation dataset.

**Figure 13 sensors-24-06039-f013:**
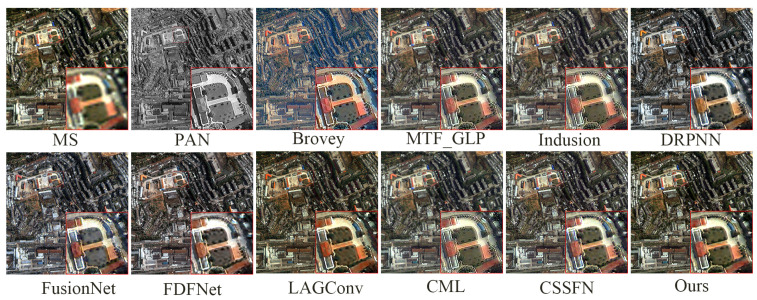
The fusion results of various methods on the QB real dataset.

**Figure 14 sensors-24-06039-f014:**
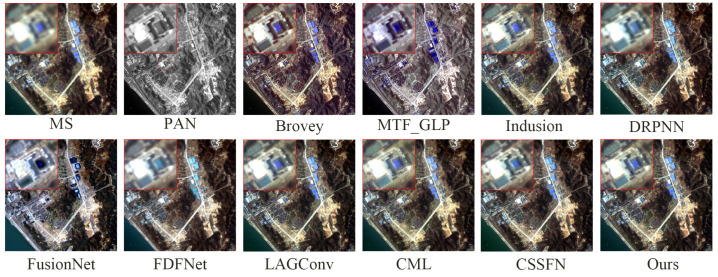
The fusion results of various methods on the GaoFen1 real dataset.

**Figure 15 sensors-24-06039-f015:**
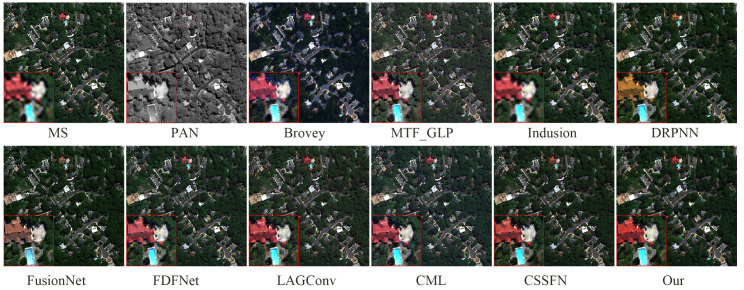
The fusion results of various methods on the WV2 real dataset.

**Figure 16 sensors-24-06039-f016:**
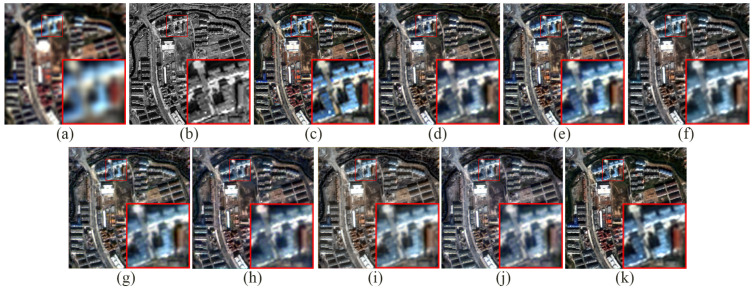
The fusion results after the removal of certain modules. (**a**) LRMS image, (**b**) PAN image, (**c**) reference image, (**d**) w/o DRRM, (**e**) w/o CFEM, (**f**) w/o SDEM, (**g**) w/o (SDEM + DRRM), (**h**) w/o (CFEM + DRRM), (**i**) w/o (CFEM + SDEM), (**j**) none_modules, and (**k**) ours.

**Figure 17 sensors-24-06039-f017:**
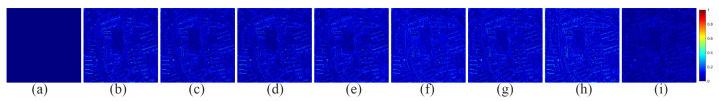
Residual plots comparing the outcomes of the experiments with different module omissions against the reference image. (**a**) Reference image, (**b**) w/o DRRM, (**c**) w/o CFEM, (**d**) w/o SDEM, (**e**) w/o (SDEM + DRRM), (**f**) w/o (CFEM + DRRM), (**g**) w/o (CFEM + SDEM), (**h**) none_modules, and (**i**) ours.

**Figure 18 sensors-24-06039-f018:**
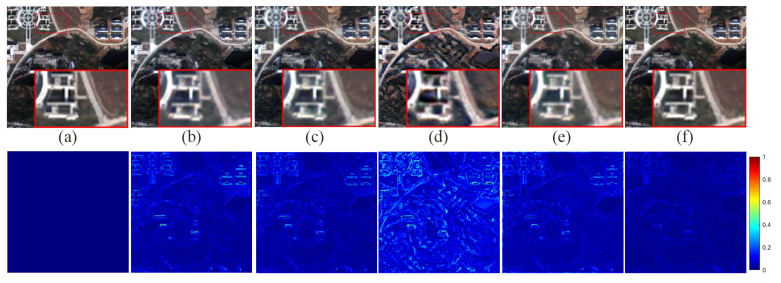
The output results of the various network structures along with their residual plots relative to the reference image. (**a**) Reference image, (**b**) concatenation, (**c**) direct injection, (**d**) one-stage, (**e**) two-stage, and (**f**) Ours.

**Table 1 sensors-24-06039-t001:** Quantitative evaluations using different methods on the simulated QB dataset.

	ERGAS	RMSE	RASE	UIQI	SAM	SCC	Q4
Brovey	1.6405	21.2350	6.3655	0.9639	1.9289	0.8976	0.5926
MTF_GLP	1.3920	17.3036	5.2954	0.9766	1.5839	0.9205	0.7221
Indusion	1.8479	23.3028	7.0379	0.9582	1.8896	0.8899	0.6299
DRPNN	1.1240	14.3054	4.3314	0.9844	1.4479	0.9413	0.7428
FusionNet	1.3804	17.3577	5.2372	0.9795	1.5534	0.9335	0.6510
FDFNet	1.0824	13.7381	4.1718	0.9858	1.3659	0.9449	0.7669
LAGConv	0.9585	12.0931	3.7216	0.98881	1.0614	0.9501	0.8065
CML	1.0689	10.8217	3.7056	0.9830	1.1712	0.9438	0.7918
CSSFN	0.8336	9.9339	3.1542	0.9888	1.1956	0.9546	0.8078
Ours	**0.6749**	**8.5607**	**2.5655**	**0.9947**	**0.8955**	**0.9801**	**0.8401**

**Table 2 sensors-24-06039-t002:** Quantitative evaluations using different methods on the simulated GF1 dataset.

	REGAS	RMSE	RASE	UIQI	SAM	SCC	Q4
Brovey	2.2433	21.9277	8.7148	0.9439	1.7472	0.8259	0.4862
MTF_GLP	6.1497	55.0688	23.0274	0.7342	7.8978	0.7393	0.2654
Indusion	2.3021	23.1673	9.0305	0.9421	1.9179	0.7842	0.5476
DRPNN	1.6585	15.7243	6.2990	0.9739	2.3238	0.8954	0.7316
FusionNet	1.8798	17.5757	7.0501	0.9669	1.6281	0.8915	0.6622
FDFNet	1.6651	16.3323	6.6479	0.9733	1.7424	0.8827	0.6628
LAGConv	1.1883	11.7103	4.6062	0.9845	1.3106	0.9169	0.8281
CML	1.4290	14.2461	5.6767	0.9769	1.6616	0.8887	0.8286
CSSFN	1.1909	10.6756	4.1047	0.9842	1.2907	0.9121	0.7768
Ours	**0.9810**	**9.4526**	**3.7265**	**0.9900**	**1.1430**	**0.9428**	**0.8644**

**Table 3 sensors-24-06039-t003:** Quantitative evaluations using different methods on the simulated WV2 dataset.

	REGAS	RMSE	RASE	UIQI	SAM	SCC	Q8
Brovey	5.6394	60.3091	19.1166	0.9197	5.1762	0.8881	0.5549
MTF_GLP	4.3400	47.1259	15.0190	0.9473	4.5941	0.9065	0.6437
Indusion	5.0325	51.9257	16.3999	0.9479	4.9926	0.8851	0.6102
DRPNN	3.6689	37.4278	12.2089	0.9612	4.0867	0.9473	0.6463
FusionNet	3.5226	35.4701	11.4256	0.9787	3.7909	0.9430	0.7746
FDFNet	4.0114	36.2872	11.7189	0.9777	3.9823	0.9444	0.7703
LAGConv	3.2990	32.6300	10.5739	0.9818	3.4230	0.9598	0.7654
CML	3.1901	31.8553	10.3129	0.9769	3.3881	0.9562	0.7831
CSSFN	3.2421	32.5700	10.5223	0.9832	3.4000	0.9546	0.7992
Ours	**2.9368**	**29.5469**	**9.5934**	**0.9851**	**3.1462**	**0.9636**	**0.8145**

**Table 4 sensors-24-06039-t004:** Quantitative evaluations using different methods on the real QB dataset.

	QNR	$D_{\lambda }$	$D_{s}$
Brovey	0.7656	0.1030	0.1514
MTF_GLP	0.8075	0.0910	0.1168
Indusion	0.8705	0.0726	0.0641
DRPNN	0.9206	0.0325	0.0496
FusionNet	0.8853	0.0643	0.0534
FDFNet	0.8845	0.0615	0.0610
LAGConv	0.9083	0.0380	0.0590
CML	0.9095	0.0453	0.0534
CSSFN	0.9192	**0.0315**	0.0474
Ours	**0.9249**	0.0320	**0.0449**

**Table 5 sensors-24-06039-t005:** Quantitative evaluations using different methods on the real GF1 dataset.

	QNR	$D_{\lambda }$	$D_{s}$
Brovey	0.6560	0.0922	0.2795
MTF_GLP	0.4245	0.2631	0.4328
Indusion	0.8900	0.0691	0.0446
DRPNN	0.8735	0.0387	0.0914
FusionNet	0.8367	0.0531	0.1174
FDFNet	0.8711	0.0348	0.0979
LAGConv	0.8954	0.0178	0.0441
CML	0.9173	**0.0107**	0.0494
CSSFN	0.8846	0.0792	0.0393
Ours	**0.9499**	0.0204	**0.0376**

**Table 6 sensors-24-06039-t006:** Quantitative evaluations using different methods on the real WV2 dataset.

	QNR	$D_{\lambda }$	$D_{s}$
Brovey	0.7332	0.1016	0.1950
MTF_GLP	0.6876	0.1583	0.2007
Indusion	0.7898	0.0747	0.1572
DRPNN	0.7974	0.0903	0.1281
FusionNet	0.8367	0.0671	0.0974
FDFNet	0.8708	0.0499	0.0702
LAGConv	0.8697	0.0366	0.0689
CML	0.8735	0.0457	0.0604
CSSFN	0.8929	0.0412	0.0620
Ours	**0.9011**	**0.0329**	**0.0593**

**Table 7 sensors-24-06039-t007:** Quantitative evaluations of ablation experiments on the simulated QB dataset.

	REGAS	RMSE	RASE	UIQI	SAM	SCC	Q4
w/o DRRM	1.4245	17.6026	5.2470	0.9822	1.9683	0.9014	0.7021
w/o CFEM	1.2024	15.4954	4.6859	0.9846	1.4687	0.9445	0.7426
w/o SDEM	1.2479	15.5463	4.6979	0.9829	1.6958	0.9418	0.7524
w/o (SDEM + DRRM)	1.4619	19.2397	5.7021	0.9755	2.0137	0.9267	0.6804
w/o (CFEM + DRRM)	1.6632	19.3664	5.7645	0.9731	1.8660	0.9343	0.5555
w/o (CFEM + SDEM)	1.3832	16.6032	5.0925	0.9828	1.7248	0.9381	0.7128
none_modules	2.2212	25.1752	7.4091	0.9625	3.1126	0.9069	0.5378
Ours	**0.6749**	**8.5607**	**2.5655**	**0.9947**	**0.8955**	**0.9801**	**0.8401**

**Table 8 sensors-24-06039-t008:** Quantitative evaluations of network structure experiments on the simulated QB dataset.

	REGAS	RMSE	RASE	UIQI	SAM	SCC	Q4
Concatenation	0.9880	12.5272	3.8471	0.9875	1.1217	0.9484	0.7996
Direct injection	0.8652	10.9196	3.3140	0.9912	1.0993	0.9673	0.8019
one_stage	4.9978	79.6737	22.52999	0.8151	1.7053	0.9258	0.6611
two_stage	1.5543	21.3100	6.2745	0.9698	0.9370	0.9776	0.8279
three_stage	**0.6749**	**8.5607**	**2.5655**	**0.9947**	**0.8955**	**0.9801**	**0.8401**

**Table 9 sensors-24-06039-t009:** Model sizes and running times of DL-based pansharpening methods.

	DRPNN	FusionNet	FDFNet	LAGConv	CML	CSSFN	Ours
Param (M)	1.64	0.15	0.09	0.03	0.16	0.35	0.07
Time (s)	6.21	5.34	3.81	9.23	5.67	6.03	5.46

## Data Availability

The data presented in this study are available on request from the corresponding author.

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
