# Peer review of "A Multi-Stage Progressive Pansharpening Network Based on Detail Injection with Redundancy Reduction"

_sensors, 2024, doi:10.3390/s24186039_

Round 1
Reviewer 1 Report
Comments and Suggestions for Authors
The paper proposes a novel method for image sharpening tasks in remote sensing imagery. The method demonstrates a certain degree of innovation, and the experiments are sufficient and logically sound. However, some minor issues need further modification. The specific comments are as follows:
1. The abstract should specify the exact accuracy improvement to highlight the effectiveness of the method.
2. The development of deep learning methods should be introduced in the introduction or related work section. Additionally, some of the latest deep learning methods should be cited to further highlight the advancement of the method. Such as Landslide extraction from aerial imagery considering context association characteristics,A cross-view intelligent person search method based on multi-feature constraints,Slice-to-slice context transfer and uncertain region calibration network for shadow detection in remote sensing imagery,Building height extraction from high-resolution single-view remote sensing images using shadow and side information.
3. Lines 105 to 124 have repetitions with the previous and subsequent issues and innovations. They can be appropriately shortened.
4. The font in Figure 1 should be enlarged for better readability by the authors, and Figure 3 is incomplete.
5. The "Parameter and Running Time Analysis" section should specify the image size and whether the experimental conditions are the same. Only then will the comparison of parameters and speed be meaningful.
Comments on the Quality of English LanguageNO
Reviewer 2 Report
Comments and Suggestions for Authors
Overall the proposed method is interesting and innovative, however, the manuscript lacks in the presentation of the details necessary for a correct reproduction of the experiments and presents a limited comparison with the state-of-the-art approaches.
Authors should provide more details about the dataset adopted for the training and the test, such as the number of images. Authors collected the dataset but no information about the cardinality of the data used is provided, nor examples of images and information about the soil content and the conditions of the images. The authors give also details about training validation and test split but it's not clear if they use the same test splits in the "real data" evaluation phase or if they use the full dataset.
Moreover, in the state of the art, multiple datasets are commonly adopted for methods training and comparison. I suggest the authors identify such datasets and adopt those for a more complete comparison.
Other minor problems regards the presentation of the method. I cannot get the meaning of section 2.2 where authors should talk about multi-stage progressive fusion, but instead compare super-resolution with pansharpening.
Figures 1 and 2 are too small and hard to read, plus Figure 3 is partially unreadable since it goes out of the page limit.
Comments on the Quality of English LanguageAuthors should perform a grammar check and revise certain parts of the text.
A few sentences are written in a strange English way. E.G. "An introductory overview of these methods is therefore warranted. "
Reviewer 3 Report
Comments and Suggestions for Authors
This paper proposes a Multi-Stage Progressive Pansharpening Network with Detail Injection with Redundancy Reduction mechanism (MSPPN- DIRRMSPN) for pansharpening. There are several problems in this article that need to be solved.
1. Please further explain and experimentally prove that the artifacts are caused by redundant information in fused results.
2. The review of existing works in the related work section is not comprehensive enough. Recently, there are many papers that use deep learning and attention mechanisms for pansharpening and feature extraction of remote sensing images.
3. It is recommended to verify the effectiveness of the proposed method in more remote sensing datasets.
4. In the ablation experiment, why is the final result of the proposed method so different from the experimental results of other rows? Is it because the corresponding methods of other rows are not trained sufficiently?
5. Figure 3 shows only part of it.
Comments on the Quality of English LanguageThis paper contains several grammatical errors. Please check carefully and correct. For example, in line 67, “For instance, A multiscale and multidepth CNN (MSDCNN)” should be “For instance, a multiscale and multidepth CNN (MSDCNN)”.
Round 2
Reviewer 3 Report
Comments and Suggestions for Authors
My concerns have been addressed.